# Prevalence, nature and predictors of omitted medication doses in mental health hospitals: A multi-centre study

Richard N. Keers[1,2,3]*, Mark Hann[4], Ghadah H. Alshehri[1], Karen Bennett[5], Joan Miller[3], Lorraine Prescott[6], Petra Brown[1,7], Darren M. Ashcroft[1,2]

1 Centre for Pharmacoepidemiology and Drug Safety, Division of Pharmacy and Optometry, School of Health Sciences, Faculty of Biology, Medicine and Health, The University of Manchester, Manchester, United Kingdom, 2 NIHR Greater Manchester Patient Safety Translational Research Centre (GM PSTRC), Manchester Academic Health Science Centre (MAHSC), The University of Manchester, Manchester, United Kingdom, 3 Greater Manchester Mental Health NHS Foundation Trust, Manchester, United Kingdom, 4 Primary Care Research Group, School of Community Based Medicine, The University of Manchester, Manchester, United Kingdom, 5 School of Health and Human Sciences, The University of Bolton, Bolton, United Kingdom, 6 North West Boroughs Health Care NHS Foundation Trust, Warrington, United Kingdom, 7 Pennine Care NHS Foundation Trust, Aston-Under-Lyne, United Kingdom

* richard.keers@manchester.ac.uk

**Data Availability Statement:** All relevant data are within the paper and its Supporting Information files.

## Abstract

### Objective

Limited evidence concerning the burden and predictors of omitted medication doses within mental health hospitals could severely limit improvement efforts in this specialist setting. This study aimed to determine the prevalence, nature and predictors of omitted medication doses affecting hospital inpatients in two English National Health Service (NHS) mental health trusts.

### Methods

Over 6 data collection days trained pharmacy teams screened inpatient prescription charts for scheduled and omitted medication doses within 27 adult and elderly wards across 9 psychiatric hospitals. Data were collected for inpatients admitted up to two weeks prior to each data collection day. Omitted doses were classified as 'time critical' and 'preventable' based on established criteria. Omitted dose frequencies were presented with 95% confidence intervals (CI). Multilevel logistic regression analyses determined the predictors of omitted dose occurrence, with omission risks presented as adjusted odds ratios (OR) with 95% CI.

### Results

18,664 scheduled medication doses were screened for 444 inpatients and 2,717 omissions were identified, resulting in a rate of 14.6% (95% CI 14.1–15.1). The rate of 'time critical' omitted doses was 19.3% (95% CI 16.3–22.6). 'Preventable' omitted doses comprised one third of all omissions (34.5%, 930/2694). Logistic regression analysis revealed that medicines affecting the central nervous system were 55% less likely to be omitted compared

**Funding:** The author(s) received no specific funding for this work. This work was supported internally by The University of Manchester and participating NHS trusts; no specific funding stream was used. These organisations had no role in the design, implementation or evaluation of this work, nor did they have a role in the decision to write and submit this manuscript for publication.

**Competing interests:** The authors have declared that no competing interests exist.

to all other medication classes (9.9% vs. 18.8%, OR 0.45 (0.40–0.52)) and that scheduled doses administered using non-oral routes were more likely to be omitted compared the oral route (inhaled OR 3.47 (2.64–4.57), topical 2.71 (2.11–3.46), 'other' 2.15 (1.19–3.90)). 'Preventable' dose omissions were more than twice as likely to occur for 'time critical' medications than non-time critical medications (50.4% vs. 33.8%, OR 2.24 (1.22–4.11)).

## Conclusions

Omitted medication doses occur commonly in mental health hospitals with 'preventable' omissions a key contributor to this burden. Important targets for remedial intervention have been identified.

## Introduction

Ensuring that medicines are managed appropriately is essential to facilitate the ongoing treatment and recovery for many patients with mental health problems [1]. Within the mental health hospital setting, there are a number of unique factors which could influence the quality and safety of medicines management processes including the health care system (e.g. presence of medicines reconciliation [2], mental health legislation [3], and drug administration practices [4, 5]), medicines used (e.g. high risk drug monitoring [6, 7], high dose/combination psychotropic prescribing [8]) and patient population (e.g. high physical health co-morbidity [9], limited insight into illness and disturbed/withdrawn behaviours [10]).

Unsafe use of medicines has recently been highlighted as a chief cause of preventable harm in health care worldwide [11], with the World Health Organisation (WHO) making understanding and improving medication-related harm a global priority in 2017 [12]. Within psychiatric hospitals, international evidence suggests that patients are frequently placed at risk from medication errors and their adverse consequences [13], with medication administration errors (MAE) one of the most common error types [13–16]. The origins of many MAEs are multifactorial, with important differences to general hospitals [10, 17].

Studies investigating the burden of MAEs in mental health hospitals consistently highlight omitted medication doses as among the most frequently observed or reported MAEs [17–19]. Omitted doses are those that are prescribed but not administered and the risk they pose to patient safety was highlighted in a patient safety alert issued in 2010 which described 183 cases of harm resulting from delayed or omitted dose incidents (n = 21,383) reported across England and Wales between September 2006 –June 2009. However, only 6.3% of the total reports originated from psychiatric settings [20].

Whilst more general studies of MAEs across mental health and general hospitals may provide data concerning the overall frequency of omitted doses, they do not contain sufficient detail to understand their nature (e.g. medications involved and underlying reasons especially unavailable drugs or 'blank boxes' not signed for administration which may be preventable) or predicting factors [13, 21]. More in-depth investigations focusing on omitted doses have helped address these needs in general hospitals including highlighting the a significant proportion of potentially 'preventable' omissions [22–24], but this data may not be generalizable to mental health hospitals and the evidence base in psychiatry is largely absent except for two single site UK studies restricted to reported missing signatures for medicines administration [25] and overall omitted dose rates within elderly care [26]. This highlights a need for further research across multiple mental health hospitals that explores in detail the type, preventability

and predictors of omissions to guide development of remedial interventions. This study therefore aimed to determine the prevalence, nature and predictors of omitted medication doses affecting inpatients in two English mental health NHS trusts.

## Methods

### Definitions

We defined an omitted medication dose as 'a dose of prescribed medication that is not administered before the next dose is due' [23]. 'Preventable' omitted doses were defined as those without a reason for omission specified on the patient's medication chart, or with a reason due to unavailability of medication at the time of administration [27]. The remaining categories of dose omission considered to be 'non-preventable' were patient refusal, omission for a clinical reason/prescriber direction, patient asleep or 'other'. 'Time critical' doses were defined as those which carry an increased risk of patient harm if a single dose is omitted, and included any of the following based on established criteria [20]: insulin, Parkinson's disease medications, anti-infective medications (excluding topical formulations), and anti-coagulants.

### Setting

This study was conducted across 9 hospitals containing 27 acute adult and elderly wards within two English mental health NHS trusts. Specialist wards such as long-stay forensics, intensive care and child and adolescent care were excluded. A total of 11 hospitals were managed by the trusts, with 1 excluded from the study as it contained long stay forensic units and the other due to local resource constraints. Both trusts utilised inpatient paper prescription and medication administration charts. The study sites each employed pharmacy teams to perform ward duties including medicines supply, medicines reconciliation and medication related advice. Medication supplies were co-ordinated utilising on-site or off-site dispensary services. Nursing staff administered medications to patients during 4 scheduled rounds (morning, lunchtime, early evening, night time). With the exception of elderly units, patients were required to attend the ward clinic/treatment room to receive their medications. Across all wards and particularly on the elderly units, 'runners' (usually trained health care assistants or nurses) were frequently utilised to bring patients to the clinic room or to take medications directly to patients for administration [4]. In order to record drug administration activity, nurses initialled a box on the prescription chart specific for particular doses, and the date/time they successfully administered it; if the dose was omitted for any reason the nurse was instructed to enter a code pertaining to the reason for the dose omission.

### Data collection

Data were collected on 6 pre-arranged data collection days between September–December 2015. The number of data collection days was chosen based on local NHS trust capacity and were scheduled to ensure data were collected over a 3 month period to better reflect usual working practices. Data collection days took place on weekdays separated by at least 2 weeks to ensure no duplication of data. Both pharmacists and pharmacy technicians could collect data. Paper inpatient prescription charts were screened for all patients on the eligible wards who were prescribed at least one regular medication and were admitted up to 14 days prior to each data collection day. This meant that a maximum of 14 days' worth of scheduled doses could be collected per patient. Patients transferred from other NHS organisations were considered new admissions, whereas internal NHS trust patient transfers were considered continuous admissions.

Data collection included all scheduled and omitted doses due at any time of day or any day of the week during each patient's screening period. Eligible medications were any listed in the British National Formulary (BNF) [28]. Information recorded included ward type (adult/ elderly), patient age/gender, medication names/forms/routes, weekend or weekday, medication round, and reason for any dose omissions. As required (*pro re nata*) medication doses were excluded, as were any medication doses scheduled when the patient was temporarily off the ward (e.g. periods of leave). Only regular and '*stat*' once only prescriptions active on the day of data collection were considered eligible for inclusion in the study; previously cancelled prescription items were excluded. Pharmacy teams examined re-written prescription charts to confirm patient eligibility and screen 14 days' worth of data.

Standardised training for pharmacist and pharmacy technician data collectors employed by the study sites involved a face-to-face seminar including an introduction to the study and guidance with completing the data collection forms. A data collection guidebook was also made available throughout the study and teams could contact the researchers with queries.

### Data analysis

Data were collated and entered into a Microsoft Excel® database by RNK, KB and GHA. Data analysis proceeded using STATA v15®. Descriptive statistics were employed to determine the point prevalence and nature of omitted doses. Rates of total omitted dose, 'preventable' omitted doses, and total/'preventable' omitted doses involving 'time critical' medications were calculated by dividing the total number of relevant omitted doses by the total number of eligible scheduled doses and were presented using frequencies and 95% confidence intervals (CI). Adjusted outcome rates were also presented for total and 'time critical' doses based on the hierarchical structure of the data.

Mixed effects logistic regression modelling was conducted to determine potential associations between total and 'preventable' omitted dose outcomes and covariates. The analyses accounted for the multi-level data structure, with doses nested within patients and patients nested within wards. Individual hospitals and NHS trusts were treated as fixed effects in the model. Given the complexity of the data structure, each potential covariate was considered, in turn, in a series of univariable models. Co-variates included ward type, day of week, medication round, administration route, medication class, 'time critical' medication, patient age and gender. Covariates with a p-value of $<0.2$ were included in a multivariable model. Non-significant ($p>0.05$) covariates were then removed, one at a time, leaving the most parsimonious model. Adjusted odds ratios (ORs) and 95% confidence intervals are presented to give context to the findings, with $p<0.05$ used to express statistical significance.

### Ethical approval

The study was approved by The University of Manchester Research Ethics Committee 5 (submission 15326) and by the audit departments at each participating NHS trust. Data were fully pseudonymised by data collectors before being sent to the research team; no patient identifiable data was accessible during analysis. Individual patient consent was not required by the research ethics committee for this retrospective service evaluation study.

## Results

A total of 18,664 scheduled medication doses were screened across 6 data collection days. These doses were intended for 444 inpatients residing on 27 wards, with the range of individual scheduled doses per ward and per patient reported as 134–1474 (mean 691) and 1–298 (mean 42), respectively.

## Total omitted doses

Pharmacy teams identified 2,717 omissions with a resulting crude omission rate of 14.6% (95% CI 14.1–15.1) of scheduled doses. After adjusting for clustering of dose omissions within individual patients, the omitted dose rate was 8.7% (95% CI 6.41–11.77%). Variability in omitted dose rates was observed across patient age, route of administration, medication round and therapeutic drug groups, with 'skin' having the highest omission rate (BNF Chapter 13, 120/314, 38.2%) and 'central nervous system' having a lower rate (9.9%, 890/8,951). There was also a large difference in omission rate between the lunchtime and night time medication rounds (20.0%, (313/1,562) vs. 12.5% (719/5,769)). The oral route of administration had the largest proportion of scheduled doses (92%) but the lowest omitted dose rate (13.3%, 2,287/17,160), with inhaled route more than double this (28.9%, 200/691). The rate of overall omissions including various subcategories is summarised in Table 1.

**'Time critical' total omissions.** Of the 18,664 scheduled doses, 637 were considered to be 'time critical'. A total of 123 of these were identified as omissions, giving a crude 'time critical' omission rate of 19.3% (95% CI 16.3–22.6%); the corresponding adjusted rate was 13.6% (95% CI 4.8–33.3%). The majority of administrations in this category were for oral medication (528/637, omission rate 21.8%; 'other' route 8/109, 7.3%). Male patients had an omission rate more than twice that of females (28.7% (86/300) vs. 11.0% (37/337)). Please see Table 2 for more details.

## 'Preventable' omitted doses

Of the 2,717 omitted doses recorded, 23 did not have a 'reason' specified leaving 2,694 which could be considered 'preventable' or 'non-preventable'. Following categorisation, a total of 34.5% (930/2694) were considered to be 'preventable'. Of these, the majority (58.1%, 540/930) were classified as 'unavailable drug'. The remaining 'non-preventable' omitted doses were most commonly patient refusal (1550/1764, 87.9%). Preventable doses are summarised in Tables 1, 2 and 3.

The topical and inhaled routes of drug administration had 'preventable' omitted dose rates more than triple that of the oral route (topical 14.7% (96/652); inhaled 15.5% (106/683); oral 4.2% (722/17,148)). The night time medication round was associated with an omission rate nearly half that of the early evening round (3.8% (219/5,764) vs. 6.9% (157/2,271)). The rate of 'time critical' 'preventable' dose omissions was twice that of 'non-time critical' omissions (9.7% (62/637) vs. 4.8% (868/18,004)). See Table 1 for more details.

**'Time critical' preventable omissions.** Half of all 'time critical' dose omissions were recorded as 'preventable' (62/123, 50.4%). Of these, the vast majority (64.5%, 40/62) were 'unavailable drug'. See Tables 2 and 3.

## Predictors of omitted doses

**Multivariate analysis–total omitted doses.** Multivariate analysis revealed that, compared to the morning medication administration round, the lunchtime (OR 1.58 (1.33, 1.88)) and early evening rounds (OR 1.24 (1.06, 1.45), both p<0.001) were associated with a statistically significant higher risk of any dose omission (there was no difference for night time rounds (OR 0.92 (0.81, 1.04)). In respect to administration route; compared to oral the topical (OR 2.71 (2.11, 3.46)), inhaled (OR 3.47 (2.64, 4.57)) and 'other' (including injectable, OR 2.15 (1.19, 3.90)) routes were all significantly associated with increased dose omission risk (all p<0.001). In contrast, the 'central nervous system' BNF chapter was associated with a 55% lower risk of omitted doses than all other chapters combined (OR 0.45 (0.40, 0.52), p<0.001). No significant associations were identified in univariate analysis for ward type, day of the

**Table 1. Crude rate of total and 'preventable' omitted doses across subcategories.**

| Category | Percentage (%) crude rate for total omitted doses (numerator / denominator) | Percentage (%) crude rate for 'preventable' omitted doses (numerator / denominator) |
|---|---|---|
| BNF1: Gastrointestinal System | 15.8% (267/1,688) | 3.9% (65/1,686) |
| BNF2: Cardiovascular System | 15.8% (347/2,198) | 6.6% (144/2,195) |
| BNF3: Respiratory System | 22.3% (197/883) | 12.9% (113/875) |
| BNF4: Central Nervous System | 9.9% (890/8,951) | 2.8% (248/8,950) |
| BNF5: Infection | 27.2% (108/397) | 14.7% (58/395) |
| BNF6: Endocrine System | 17.4% (158/910) | 5.5% (50/910) |
| BNF7: Genito-urinary System | 19.5% (30/154) | 10.4% (16/154) |
| BNF8: Immune / Malignancy | 3.3% (1/30) | 3.3% (1/30) |
| BNF9: Blood and Nutrition | 18.7% (513/2,744) | 5.2% (143/2,738) |
| BNF10: Musculoskeletal | 8.3% (8/97) | 5.2% (5/97) |
| BNF11: Eye | 25.9% (42/162) | 9.3% (15/162) |
| BNF12: Ear, Nose, Oropharynx | 26.5% (36/136) | 17.8% (24/135) |
| BNF13: Skin | 38.2% (120/314) | 15.3% (48/314) |
| **Total** | **18,664** | **18,641**[*] |
| Oral route | 13.3% (2,287/17,160) | 4.2% (722/17,148) |
| Topical route | 28.5% (224/786) | 14.7% (96/652) |
| Inhaled route | 28.9% (200/691) | 15.5% (106/683) |
| Other route** | 22.2% (6/27) | 3.8% (6/158) |
| **Total** | **18,664** | **18,641**[*] |
| Morning round | 14.5% (1,315/9,058) | 5.0% (452/9,045) |
| Lunchtime round | 20.0% (313/1,562) | 6.5% (102/1,561) |
| Early evening round | 16.3% (370/2,275) | 6.9% (157/2,271) |
| Bedtime round | 12.5% (719/5,769) | 3.8% (219/5,764) |
| **Total** | **18,664** | **18,641**[*] |
| Adult ward | 13.9% (1,935/13,931) | 5.2% (719/13,913) |
| Elderly ward | 16.5% (782/4,733) | 4.5% (211/4,728) |
| **Total** | **18,664** | **18,641**[*] |
| Male patients | 11.5% (955/8,290) | 4.0% (335/8,277) |
| Female patients | 17.0% (1,762/10,374) | 5.7% (595/10,364) |
| **Total** | **18,664** | **18,641**[*] |
| Patient age between 18–34 | 14.8% (489/3,316) | 5.6% (184/3,312) |
| Patient age between 35–44 | 9.8% (387/3,935) | 3.7% (145/3,933) |
| Patient age between 45–54 | 15.3% (487/3,175) | 5.7% (181/3,169) |
| Patient age between 55–64 | 13.3% (265/1,987) | 6.2% (124/1,987) |
| Patient age between 65–74 | 15.4% (520/3,369) | 3.6% (121/3,361) |
| Patient age between 75–94 | 19.7% (569/2,882) | 6.1% (175/2,879) |
| **Total** | **18,664** | **18,641**[*] |
| Weekday doses | 14.7% (2,125/14,436) | 4.8% (698/14,421) |
| Weekend doses | 14.0% (592/4,228) | 5.5% (232/4,220) |
| **Total** | **18,664** | **18,641**[*] |
| 'Time critical' dose | 19.3% (123/637) | 9.7% (62/637) |
| 'Non time critical' dose | 14.3% (2,571/18,027) | 4.8% (868/18,004) |

*(Continued)*

**Table 1.** (Continued)

| Category | Percentage (%) crude rate for total omitted doses (numerator / denominator) | Percentage (%) crude rate for 'preventable' omitted doses (numerator / denominator) |
|---|---|---|
| Total | 18,664 | 18,641* |

BNF: British National Formulary [28] * Excludes 23 doses without reason for omission specified ** Includes injections

week, patient age, patient gender and 'time critical' medications were not taken forward to multi-variate analysis. Details of the predictor analysis for total omitted doses are presented in Table 4.

**Multivariate analysis–'preventable' omitted doses.** Compared with 'non-preventable' omissions, 'preventable' omitted doses were reported to be twice as likely to occur with 'time critical' doses than 'non-time critical' doses (OR 2.24 (1.22–4.11), $p = 0.01$). 'Preventable' dose omissions were 46% more likely to occur than 'non-preventable' omissions for the early evening medication round (OR 1.46 (1.01–2.12), $p < 0.001$, compared to morning) and were 37%

**Table 2. Crude rate of 'time critical' medication omissions across subcategories.**

| Category | Percentage (%) crude rate (numerator & denominator) | |
|---|---|---|
| | **Total omitted doses** | **'Preventable' omitted doses** |
| Oral route | 21.8% (115/528) | 50.4% (58/115) |
| Topical route | 0% (0/0) | 0% (0/0) |
| Inhaled route | 0% (0/0) | 0% (0/0) |
| Other route* | 7.3% (8/109) | 50.0% (4/8) |
| Total | 637 | 123** |
| Morning round | 20.2% (58/287) | 51.7% (30/58) |
| Lunchtime round | 31.0% (22/71) | 63.6% (14/22) |
| Early evening round | 12.1% (17/141) | 47.1% (8/17) |
| Bedtime round | 18.8% (26/138) | 38.5% (10/26) |
| Total | 637 | 123** |
| Adult ward | 19.1% (82/430) | 68.3% (56/82) |
| Elderly ward | 19.8% (41/207) | 14.6% (6/41) |
| Total | 637 | 123** |
| Male patients | 28.7% (86/300) | 50.0% (43/86) |
| Female patients | 11.0% (37/337) | 51.4% (19/37) |
| Total | 637 | 123** |
| Patient age between 18–34 | 31.6% (43/136) | 67.4% (29/43) |
| Patient age between 35–44 | 3.5% (6/171) | 100% (6/6) |
| Patient age between 45–54 | 46.4% (26/56) | 76.9% (20/26) |
| Patient age between 55–64 | 4.8% (1/21) | 0% (0/1) |
| Patient age between 65–74 | 16.9% (20/118) | 10.0% (2/20) |
| Patient age between 75–94 | 20.0% (27/135) | 18.5% (5/27) |
| Total | 637 | 123** |
| Weekday doses | 20.8% (104/499) | 51.9% (54/104) |
| Weekend doses | 13.8% (19/138) | 42.1% (8/19) |
| Total | 637 | 123** |

* Includes: injectable medication

** Excludes doses without a reason specified

**Table 3. Frequency of omitted dose reasons by category.**

| Category | Frequency (%) | | | | | | Total |
|---|---|---|---|---|---|---|---|
| | 'Non-preventable' omitted doses | | | | 'Preventable' omitted doses | | |
| | Clinical reason | Patient refusal | Patient asleep | Other | Unavailable drug | Not signed | |
| All omitted doses | 100 (3.7) | 1550 (57.5) | 70 (2.6) | 44 (1.6) | 540 (20.0) | 390 (14.5) | 2694 (100) |
| 'Time critical' omitted doses | 2 (1.6) | 55 (44.7) | 3 (2.4) | 1 (0.8) | 40 (32.5) | 22 (17.9) | 123 (100) |

less likely for the night time medication round (OR 0.63 (0.46–0.86), p<0.001). The inhaled route of administration (OR 1.83 (1.06–3.16), compared to oral) was found to be more likely to be associated with 'preventable' rather than 'non-preventable' dose omissions (p<0.01), with the 'other' administration route 85% less likely to be associated with 'preventable' than 'non-preventable' omissions (OR 0.15 (0.04–0.58), p = 0.006). Patients residing on elderly wards were 74% less likely to experience a 'preventable' omission compared to a 'non-preventable' omission compared to those on adult wards (OR 0.26 (0.08–0.79), p = 0.018), and 'preventable' omissions were more than three times more likely on weekends compared to

**Table 4. Univariable and Multivariable Associations from Logistic Regression Analyses of Total Omitted Doses on Patient-, Drug/ Dose- and Ward-Specific Covariates.**

| Covariate | Category | Odds Ratios for Associations between Omitted Dose and Covariates | | | |
|---|---|---|---|---|---|
| | | Univariable | | Multivariable | |
| | | Coefficient (95% C.I.) | p-value | Coefficient (95% C.I.) | p-value |
| Ward Type | Adult | Reference | 0.025 | | NS |
| | Elderly | 2.01 (1.09, 3.71) | | | |
| Day of the Week | Weekday | Reference | 0.215 | | NS |
| | Weekend | 1.08 (0.96, 1.22) | | | |
| Medicines Administration Round | Morning | Reference | <0.001 | Reference | <0.001 |
| | Lunchtime | 1.49 (1.26, 1.77) | | 1.58 (1.33, 1.88) | |
| | Early Evening | 1.24 (1.06, 1.45) | | 1.24 (1.06, 1.45) | |
| | Night Time | 0.82 (0.72, 0.92) | | 0.92 (0.81, 1.04) | |
| Administration Route | Oral | Reference | <0.001 | Reference | <0.001 |
| | Inhaled | 4.60 (3.53, 5.99) | | 3.47 (2.64, 4.57) | |
| | Topical | 3.77 (2.97, 4.79) | | 2.71 (2.11, 3.46) | |
| | Other | 2.71 (1.52, 4.85) | | 2.15 (1.19, 3.90) | |
| Medication Class | Non-Central Nervous System | Reference | <0.001 | Reference | <0.001 |
| | Central Nervous System | 0.38 (0.33, 0.42) | | 0.45 (0.40, 0.52) | |
| Time Critical Medication | Not Time Critical | Reference | 0.005 | | NS |
| | Time Critical | 1.56 (1.14, 2.13) | | | |
| Patient Age-Group | 18–34 | Reference | 0.001 | | NS |
| | 35–44 | 0.78 (0.42, 1.44) | | | |
| | 45–54 | 1.41 (0.74, 2.69) | | | |
| | 55–64 | 1.71 (0.82, 3.57) | | | |
| | 65–74 | 1.87 (0.91, 3.85) | | | |
| | 75–94 | 4.04 (1.93, 8.44) | | | |
| Patient Gender | Male | Reference | 0.116 | | NS |
| | Female | 0.71 (0.46, 1.09) | | | |

NS. Covariate was not statistically significant at the 5% level (i.e. p>0.05).

**Table 5. Univariable and Multivariable Associations from Logistic Regression Analyses of 'Preventable' Omitted Doses on Patient-, Drug/ Dose- and Ward-Specific Covariates.**

| Covariate | Category | Odds Ratios for Associations between 'Preventable' Omitted Dose and Covariates | | | |
|---|---|---|---|---|---|
| | | Univariable | | Multivariable | |
| | | Coefficient (95% C.I.) | p-value | Coefficient (95% C.I.) | p-value |
| Ward Type | Adult | Reference | 0.028 | Reference | 0.018 |
| | Elderly | 0.30 (0.10, 0.88) | | 0.26 (0.08, 0.79) | |
| Day of the Week | Weekday | Reference | <0.001 | Reference | <0.001 |
| | Weekend | 3.47 (2.53, 4.74) | | 3.44 (2.51, 4.72) | |
| Medicines Administration Round | Morning | Reference | <0.001 | Reference | <0.001 |
| | Lunchtime | 1.10 (0.75, 1.63) | | 1.05 (0.71, 1.57) | |
| | Early Evening | 1.48 (1.03, 2.13) | | 1.46 (1.01, 2.12) | |
| | Night Time | 0.65 (0.48, 0.88) | | 0.63 (0.46, 0.86) | |
| Administration Route | Oral | Reference | 0.057 | Reference | 0.006 |
| | Inhaled | 1.51 (0.90, 2.55) | | 1.83 (1.06, 3.16) | |
| | Topical | 1.06 (0.63, 1.77) | | 1.03 (0.60, 1.76) | |
| | Other | 0.26 (0.08, 0.88) | | 0.15 (0.04, 0.58) | |
| Medication Class | Non-Central Nervous System | Reference | 0.111 | | NS |
| | Central Nervous System | 0.78 (0.58, 1.06) | | | |
| Time Critical Medication | Not Time Critical | Reference | 0.065 | Reference | 0.010 |
| | Time Critical | 1.69 (0.97, 2.96) | | 2.24 (1.22, 4.11) | |
| Patient Age-Group | 18–34 | Reference | 0.981 | | NS |
| | 35–44 | 0.88 (0.25, 3.02) | | | |
| | 45–54 | 0.65 (0.18, 2.34) | | | |
| | 55–64 | 0.84 (0.21, 3.38) | | | |
| | 65–74 | 0.68 (0.17, 2.65) | | | |
| | 75–94 | 0.65 (0.17, 2.48) | | | |
| Patient Gender | Male | Reference | 0.377 | | NS |
| | Female | | | | |

NS. Covariate was not statistically significant at the 5% level (i.e. p>0.05).

weekdays (OR 3.44 (2.51–4.72), p<0.001). The topical route and lunchtime medication rounds showed no significant association in multivariate analysis, and patient age, gender and medication class did not progress from univariate modelling. For full details of predictor analysis please see Table 5.

## Discussion

This is the first study to explore the prevalence, nature and predictors of omitted doses of medication across multiple mental health hospitals. Our findings suggest that omitted doses are common in this setting with a third considered to be 'preventable'. We found evidence that doses for medications designed to treat conditions outside the central nervous system (CNS) appeared more likely to be omitted than those for CNS drugs. 'Preventable' omissions were twice as likely to affect 'time critical' than non-'time critical' doses, affected patients residing on elderly wards less frequently and were three times as likely to occur on weekends. Non-oral routes of administration also emerged as an important predictor for omitted doses across both overall and 'preventable' omitted doses.

## Implications of findings

Medication errors are currently of global interest, with recent evidence highlighting their major role in causing preventable harm in health care [11]. Our findings help respond to the WHO Global Patient Safety Challenge 'Medication Without Harm' agenda by identifying medication, patient and system factors associated with omitted dose and in particular 'preventable' omitted dose risk, which informs targeted interventions under the 'high-risk situations' priority for action [12].

Limited progress has been made in mental health settings to routinely monitor omitted doses, openly share and benchmark data, and then work towards reducing their burden as part of quality improvement efforts [26, 29]. Our study findings support recommendations [30, 31] for increased attention to using meaningful, routinely collected and accessible data such as this as part of quality improvement efforts, with a focus on 'preventable' omissions to drive local improvement efforts.

'Preventable' dose omissions have emerged as an important target for remedial intervention in this research. The ratio of 'preventable' to 'non-preventable' omissions that we observed appears broadly similar to data from general hospital studies [23, 24, 27]. Available evidence for the causes of unavailable drug related 'preventable' omitted doses from mental health hospitals [14–16] identifies causative factors such as medicines logistics, but detail is often limited to codes on prescription charts or brief descriptions in incident reports. Further in-depth investigation is therefore required as seen elsewhere for general MAEs [10] to help inform interventions tailored to the mental health setting.

Our finding that medications within the CNS class (containing all psychotropic drugs) had a lower risk of being omitted than those from other classes was perhaps not surprising given the specialist psychiatric setting for this study. Indeed, a greater proportion of non-psychotropic drugs were found to be affected by MAEs than psychotropics in one UK based MAE study on two long stay older person psychiatric wards [18]. However, this relationship was not present for 'preventable' omissions compared to 'non-preventable' and indicates that health providers could focus attention on supporting patients and staff to reduce refusal of medication in clinical practice as this was the most common reason for 'non-preventable' omissions. A narrative literature review study published in 2011 helps to identify the implications and factors associated with inpatient medication refusal in psychiatry to guide local improvement efforts, but further research is required to explore the aetiology of dose refusals to inform this activity [32].

Medication doses administered using non-oral routes were consistently associated with at least two fold risk of overall (and 'preventable' for inhaled doses) omissions compared to the oral administration route. One other published study of MAEs in mental health hospitals conducted a similar comparison, finding that a greater proportion of errors involved non-oral administration routes on two elderly units [18]. Targeted investigation into the causes of this observation could include exploration of variable practices of medication storage (noted to be a causative factor in some types of MAE in psychiatry [9]) and awareness and training of specialist mental health staff in physical health illnesses which has been previously highlighted as an area for development for nurses, for example [33, 34].

Our finding that the rate of omissions of 'time critical' medication doses was higher than overall scheduled doses, and that these omissions were twice as likely to affect 'preventable' omitted doses compared to non-time critical doses is concerning in light of the high risk of patient harm these medications are known to pose, with a UK national alert issued 10 years ago [20]. This should instigate renewed and prompt action within wider mental health care organisations to thoroughly and routinely investigate the burden and causes of this issue in

order to develop effective and sustainable solutions. In one example, researchers from Australia described the successful development and implementation of a 'time-critical' medication identification and audit tool in 11 hospitals, with staff finding the tool useful to inform improvement efforts [35]. When considered alongside our finding that 'preventable' omissions were three times more likely to occur at weekends than weekdays compared to 'non-preventable' omissions, this highlights the need for health providers to review medicines supply procedures at weekends such as provision of emergency drug cupboards and input of pharmacy services. In recent years, general hospitals in the UK have seen the introduction of twenty-four hour pharmacy services in order to improve safety including dose omissions [36, 37].

As one of the most commonly occurring MAEs, omitted doses have been the subject of improvement interventions in general hospitals targeting pharmacy staff/systems [27, 38], nursing education, information technology and error reporting schemes [39], some with mixed results. The evidence base in psychiatry requires expansion as it is limited to a national UK benchmarking initiative [29], and two positive single site studies of awareness/benchmarking [26] and automated dispensing cabinets [40]. The introduction of electronic prescribing and medication administration (EPMA) systems may be expected to reduce certain omitted doses (e.g. prescription not signed for administration), and the two participating mental health trusts in this study are currently working towards implementation. However, evidence from general hospitals indicates that 'preventable' omissions may persist despite the use of EPMA [41, 42].

## Strengths and limitations

Important strengths of this study include data collection taking place across 9 hospitals within two large NHS trusts, the use of standardised training across participating sites, adoption of a design which minimised double counting of scheduled/omitted doses and the exploration of a number of risk factors for both overall and 'preventable' omitted doses. However, it is impossible to rule out variations in data collection between pharmacy teams, and our study sites were confined to one geographical region in England which could limit generalisability.

Our data collection process was designed to balance the retrieval of optimal omitted dose data against limited pharmacy team capacity. As such, we were not able to collect data beyond 2 weeks' hospital stay for each eligible patient, nor were we able to collect data on any medications that may have been prescribed during this period but were not 'active' on the data collection day. In order to maximise generalisability of our findings, we excluded specialist wards such as intensive care and child/adolescent care and future work should determine if omitted dose rates differ on these units. Whilst we could not determine the actual/potential severity of recorded omitted doses, by including 'time critical' medications we were able to assign clinical meaning to our findings. Whilst some may not include certain types of omissions such as doses refused/omitted for clinical reasons as omitted doses [43], they were included in this study and we separated data for 'preventable' omitted doses. It is also theoretically possible for some omitted doses considered to be 'non-preventable' to actually be 'preventable' (e.g. patient refuses due to correctable lack of understanding of medication). However, confirmation would require extensive investigation in the clinical setting and was beyond the resource capabilities for many data collection teams.

## Conclusion

This is the first in-depth exploration of the prevalence, nature and predictors of omitted doses in mental health hospitals. Omissions were recorded for approximately 1 in 7 scheduled doses, with similar numbers of 'time critical' doses affected. 'Preventable' omitted doses emerged as

an important target for remedial intervention, accounting for more than one third of overall omitted doses and being twice as likely to affect 'time critical' doses than 'non-time critical' doses. The findings of this study should be used to inform the development of future research and quality improvement interventions designed to reduce the burden of omitted doses in psychiatric hospitals.

## Supporting information

**S1 File. Final Data Set—Omitted Dose Study (RKeers).**
(XLS)

## Acknowledgments

We would like to thank all the pharmacy teams at each participating NHS trust site for their role in data collection, and particularly the site leads for delivering study training and for helping to manage the data collection process.

## Author Contributions

**Conceptualization:** Richard N. Keers, Darren M. Ashcroft.

**Data curation:** Richard N. Keers, Mark Hann, Ghadah H. Alshehri, Karen Bennett.

**Formal analysis:** Mark Hann.

**Methodology:** Richard N. Keers, Darren M. Ashcroft.

**Project administration:** Richard N. Keers, Karen Bennett, Joan Miller, Lorraine Prescott, Petra Brown, Darren M. Ashcroft.

**Supervision:** Richard N. Keers, Karen Bennett, Joan Miller, Lorraine Prescott, Petra Brown, Darren M. Ashcroft.

**Validation:** Richard N. Keers, Ghadah H. Alshehri, Karen Bennett.

**Writing – original draft:** Richard N. Keers.

**Writing – review & editing:** Richard N. Keers, Mark Hann, Ghadah H. Alshehri, Karen Bennett, Joan Miller, Lorraine Prescott, Petra Brown, Darren M. Ashcroft.

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
