## [Decision Letter · Decision Letter 0]

15 Nov 2019

PONE-D-19-23592

Prevalence, nature and predictors of omitted medication doses in mental health hospitals: a multi-centre study

PLOS ONE

Dear Dr. Keers,

Thank you for submitting your manuscript to PLOS ONE. After careful consideration, we feel that it has merit but does not fully meet PLOS ONE’s publication criteria as it currently stands. Therefore, we invite you to submit a revised version of the manuscript that addresses the points raised during the review process.

We would appreciate receiving your revised manuscript by Dec 30 2019 11:59PM. To enhance the reproducibility of your results, we recommend that if applicable you deposit your laboratory protocols in protocols.io, where a protocol can be assigned its own identifier (DOI) such that it can be cited independently in the future. For instructions see: http://journals.plos.org/plosone/s/submission-guidelines#loc-laboratory-protocols

We look forward to receiving your revised manuscript.

Kind regards,

Prof, Mojtaba Vaismoradi, PhD, MScN, BScN

Academic Editor

PLOS ONE

2.  In ethics statement in the manuscript and in the online submission form, please provide additional information about the patient records used in your retrospective study. Specifically, please ensure that you have discussed whether all data were fully anonymized before you accessed them and/or whether the IRB or ethics committee waived the requirement for informed consent. If patients provided informed written consent to have data from their medical records used in research, please include this information.

Reviewers' comments:

Reviewer #1: The paper is very well written and the subject is also considerable

Please also consider the following:

Please mention how 9 hospitals were selected? Did you get all the hospitals in the two centers?

How did you calculate the sample size needed for this study? How was the sampling done?

Were patients also satisfied with the use of medical information?

Reviewer #2: The structure of the manuscript is well put together, however it has some defects that listed in below:

Results

- Line 179: “The rate of overall omissions including various subcategories is summarised in Table 1”. Why did you report total omitted doses including, ‘preventable’ omitted doses? If it is total, what about a column for ‘non-preventable’?

- Please list out your exact findings related to predictors factors and then discuss about them. I cannot find them easily in the text, as a reader.

Ethical approval

- Further information is needed. For instance, any permission to access inpatient prescription charts?

Discussion

- Line 250: “We found evidence that particular patients may be experiencing more omitted doses”. What do you mean from "particular patients"?

- Discussion more about implications of findings in clinical practice.

---

## [Author Response · Author response to Decision Letter 0]

7 Jan 2020

RESPONSE TO REVIEWER COMMENTS 

PONE-D-19-23592 ‘Prevalence, nature and predictors of omitted medication doses in mental health hospitals: a multi-centre study’

Please see the attached file for the response to reviewer comments (formatted)

Editorial Comments:

Acknowledged. 

2. In ethics statement in the manuscript and in the online submission form, please provide additional information about the patient records used in your retrospective study. Specifically, please ensure that you have discussed whether all data were fully anonymized before you accessed them and/or whether the IRB or ethics committee waived the requirement for informed consent. If patients provided informed written consent to have data from their medical records used in research, please include this information.

 Acknowledged – we have now made clear in the ethics statement that data were fully pseudonymised before being sent to the research team for analysis and that individual patient consent was not required by the research ethics committee for this service evaluation study. 

Reviewers' comments:

Reviewer #1: The paper is very well written and the subject is also considerable

Many thanks for these kind words and for reviewing our manuscript. 

Please also consider the following:

Please mention how 9 hospitals were selected? Did you get all the hospitals in the two centers?

A total of 11 hospitals were covered by the 2 participating NHS trusts, with 1 being excluded as this focused on long-stay forensic wards rather than acute ward environments and another due to reasons of local resource capacity. Specialist wards such as long-stay forensics, intensive care and child and adolescent care were excluded in order to help ensure study findings were generalizable. The manuscript (page 6) and our ‘Strengths and Limitations’ section in the Discussion (page 20) has now been updated to reflect this point. 

How did you calculate the sample size needed for this study? How was the sampling done?

As this study was an exploratory service evaluation to measure medicines administration quality we did not perform a sample size ‘power’ calculation. The number of data collection days (n=6) was chosen based on local capacity as each NHS trust provided the data collectors (we also refer to NHS team capacity in our ‘Strengths and Limitations’ section in the Discussion), and the days were scheduled to ensure data were collected over a 3 month period to better reflect usual working practices. All patients on each ward were included if eligible following screening. The manuscript has now been updated with this information (see page 7). 

Were patients also satisfied with the use of medical information?

As per yours and additional editorial comments, we have added a statement to our ‘Ethical Approval’ section which states that individual patient consent was not required by the research ethics committee for this service evaluation study.

Reviewer #2: The structure of the manuscript is well put together, however it has some defects that listed in below:

Results

- Line 179: “The rate of overall omissions including various subcategories is summarised in Table 1”. Why did you report total omitted doses including, ‘preventable’ omitted doses? If it is total, what about a column for ‘non-preventable’?

Many thanks for reviewing our manuscript and for providing these valuable comments.

An important aim of our work was to present both the overall omitted dose rates, and the ‘preventable’ rates to the reader as findings concerning these ‘preventable’ omissions may help better focus local quality improvement efforts and future research in the field. However, we acknowledge that our focus on preventability of omissions could be clearer in our introduction and we have added text to achieve this (page 5) as well as in the discussion (page 17).

We present the overall numbers of ‘preventable’ and ‘non-preventable’ omissions in Table 3 so the reader can compare their relative burden, and by comparing the numerator figures across columns 1 and 2 in Table 1, the reader can also determine the rates for ‘non-preventable’ omissions. This helps to focus attention on ‘overall’ and ‘preventable’ omissions whilst avoiding making Table 1 larger and more difficult to process for the reader. 

- Please list out your exact findings related to predictors factors and then discuss about them. I cannot find them easily in the text, as a reader.

Within the results section page 14, we have a section entitled ‘Predictors of omitted doses’ where we describe all the statistically significant findings reported in Tables 4 and 5. We have now expanded on this section to include all findings (whether significant or not) from the analysis. 

Ethical approval

- Further information is needed. For instance, any permission to access inpatient prescription charts?

As per yours and additional editorial comments, we have added a statement to our ‘Ethical Approval’ section which states that individual patient consent was not required by the research ethics committee for this service evaluation study.

Discussion

- Line 250: “We found evidence that particular patients may be experiencing more omitted doses”. What do you mean from "particular patients"?

Many thanks for this comment. We have removed reference to ‘particular patients’ and have added additional details concerning the predictors for ‘preventable’ omissions in this paragraph.

- Discussion more about implications of findings in clinical practice.

Based on your earlier helpful comments we have added some detail to the Discussion concerning the implications for preventable omissions. We have also now reviewed the Discussion and added lines of text across sections to help better link the findings to clinical practice, including ‘preventable’ omissions occurring at weekends and addressing patient refusals as a driving factor behind non-preventable omissions.

---

## [Decision Letter · Decision Letter 1]

27 Jan 2020

Prevalence, nature and predictors of omitted medication doses in mental health hospitals: a multi-centre study

PONE-D-19-23592R1

Dear Dr. Keers,

We are pleased to inform you that your manuscript has been judged scientifically suitable for publication and will be formally accepted for publication once it complies with all outstanding technical requirements.

With kind regards,

Prof, Mojtaba Vaismoradi, PhD, MScN, BScN

Academic Editor

PLOS ONE

Reviewers' comments

Reviewer #1: (No Response)

Reviewer #2: Dear authors, Thank you for addressing all comments sufficiently. I therefore recommend to accept the manuscript for publication.

---

## [Editor Report · Acceptance letter]

29 Jan 2020

PONE-D-19-23592R1 

Prevalence, nature and predictors of omitted medication doses in mental health hospitals: a multi-centre study 

Dear Dr. Keers:

I am pleased to inform you that your manuscript has been deemed suitable for publication in PLOS ONE. Congratulations! Your manuscript is now with our production department. 

With kind regards,

on behalf of

Professor Mojtaba Vaismoradi 

Academic Editor

PLOS ONE